# Beyond Monoculture: A Comparative Analysis of Soil Properties and Grain Quality in Rice-Based Co-Culture Systems

**DOI:** 10.3390/biology14091195

**Published:** 2025-09-04

**Authors:** Yang Xu, Geye Ding, Weiwei Ma, Jiao Yuan, Jing Liu, Ziyu Xie, Junde Guo, Linzhi Ou, Huang Huang, Can Chen, Junhua Li

**Affiliations:** 1College of Fisheries, Hunan Agricultural University, Changsha 410128, China; xy_961112@163.com (Y.X.); 15057259561@163.com (G.D.); ljj5035@163.com (J.L.); 17779550759@163.com (Z.X.); 2Yueyang Agricultural and Rural Affairs Centre, Yueyang 414000, China; mww20190901@163.com; 3College of Agronomy, Hunan Agricultural University, Changsha 410128, China; 13548752603@139.com (J.Y.); hh8631@hunau.net (H.H.); 4Hunan Engineering Research Center of Rice Field Ecological Planting and Breeding, Changsha 410128, China; 5Agricultural Comprehensive Service Center of Yiyang Sub-District, Hengyang 421500, China; pederguo@126.com; 6Hunan Yahua Seed Industry Co., Ltd., Changsha 410128, China; m15200680286@163.com

**Keywords:** rice-chicken-fish co-culture, soil fertility dynamics, soil enzymatic activity, soil microbial community, rice grain quality

## Abstract

Conventional rice monoculture degrades soil health through intensive tillage and chemical inputs, threatening sustainable production. This study systematically evaluated stage-specific impacts of rice-based ecological co-culture systems (rice-chicken, rice-fish, and rice-chicken-fish) on soil fertility dynamics, enzymatic activity, microbial communities, and grain quality relative to conventional tillage. It sought to elucidate mechanisms linking soil–microbial interactions to quality enhancement. Findings revealed that co-culture systems significantly modified soil properties in growth stage-dependent patterns, with rice-chicken-fish enhancing late-season organic matter accumulation, while rice-chicken consistently improved potassium availability. Soil enzyme activities exhibited distinct temporal responses, including delayed urease peaks in fish-integrated systems. Microbial communities underwent restructuring with increased network complexity in co-cultures despite conserved alpha diversity. Grain quality improved system-specifically, with rice-chicken enhancing milling recovery and reducing chalkiness, whereas rice-fish and rice-chicken-fish increased protein content. Integrative analysis linked soil pH, organic matter, invertase, and specific microbial taxa (e.g., *Nitrospira* and *Syntrophus*) to grain quality attributes. Collectively, ecological co-culture systems enhanced agroecosystem multifunctionality by optimizing soil–plant–microbe interactions, improving nutrient mobilization, microbial resilience, and targeted grain quality dimensions. This work provided a science-based framework for designing precision agroecological practices that reduce synthetic inputs while improving rice productivity and nutritional quality, thereby supporting sustainable food security.

## 1. Introduction

Amidst growing global population pressures and diminishing arable land, ensuring food security has emerged as an urgent priority [1]. Global rice (*Oryza sativa* L.) cultivation faces escalating challenges in achieving sustainable intensification, necessitating yield stability while countering issues such as depleted soil fertility, increasing water scarcity, and ongoing environmental degradation [2]. Conventional paddy monocultures, characterized by intensive tillage and chemical inputs, accelerate soil organic matter depletion and disrupt microbial ecosystems, triggering compensatory dependency on synthetic fertilizers that further compromise nutrient cycling and long-term agricultural resilience [3,4]. Ecological co-culture systems integrating aquatic species and/or poultry have emerged as promising alternatives, leveraging biodiversity to enhance nutrient cycling, pest control, and resource efficiency [5,6,7,8,9]. However, these systems also have certain limitations, such as environmental pressures, potential trade-offs in resource allocation, and management complexity.

Soil fertility constitutes the fundamental pillar of agricultural productivity, encompassing physicochemical properties and biological components that collectively govern crop growth, stress resilience, and yield quality [10]. Beyond serving as nutrient reservoirs, fertile soils function as dynamic matrices where microbiota, as biological engines driving nutrient transformations, orchestrate essential nutrient transformations through enzymatic processes [11]. Specifically, microbial communities mineralize organic nitrogen [12], solubilize phosphorus [12], regulate phytohormone signaling [13], optimize soil structure [11], and thereby modulate plant health. Critically, microbial metabolic architecture directly influences root exudation profiles, nutrient-use efficiency, photosynthetic partitioning, and systemic-acquired resistance, thereby transforming crops from passive nutrient recipients into active participants within a symbiotic soil-plant continuum [14].

Ecological co-culture systems reshape this continuum through multifaceted mechanisms, such as physicochemical modulation, nutrient cycling augmentation, and microbial community restructuring. Faunal bioturbation, like foraging, swimming, and burrowing, improve soil porosity and aggregation, enhancing oxygen diffusion and organic matter distribution [6,15,16]. Meta-analysis confirmed that co-culture systems of rice and aquatic animals elevate soil organic carbon content by 11.6% via optimized nutrient mobilization [6]. Empirical studies demonstrated that rice-crayfish systems increase microbial necromass contributions to soil nitrogen pools by 18–27% through accelerated microbial turnover [7], while rice-duck co-culture enhanced organic matter accumulation, potassium availability, and alkali-hydrolyzable nitrogen (AHN) content [8].

Microbiome reconfiguration yields particularly significant benefits. Rice-prawn co-culture elevated microbial diversity while restructuring community composition, particularly reducing Proteobacteria-associated pathogens and increasing Chloroflexi abundance for improved nutrient cycling, thereby fostering a more modular and ecologically resilient soil microbial community [17]. Integrated systems (e.g., rice-duck-crayfish) further boosted humus content and microbial enzyme activities [9]. Notably, while rice-crayfish systems elevate the potential dissemination risk of antibiotic resistance genes (ARGs) [18], long-term rice-eel co-culture suppressed ARG accumulation and mobility [19].

Rice grain quality encompasses a suite of interdependent physicochemical and sensory attributes defined by four critical domains including processing quality, appearance quality, cooking/eating quality, and nutritional quality [20,21]. While fundamentally influenced by genotype, environmental drivers, and agronomic management practices, ecological co-culture systems primarily modulate grain quality through strategic modification of the agroecological environment [22]. Empirical evidence demonstrated that integrated rice-duck-crayfish systems significantly elevate protein content (8.15%) and gel consistency (6.52%) while simultaneously improving milling recovery rates and reducing chalky rice rate and chalkiness degree [9].

Despite potential for enhancing agroecosystem multifunctionality, critical knowledge gaps persist regarding the stage-specific mechanisms through which different rice-based co-culture systems modulate soil fertility, enzymatic activities, microbial communities, and their integrated effects on grain quality. Previous studies have often focused on single-system evaluations or limited growth stages, lacking a comparative and temporally resolved analysis across multiple co-culture regimes. Moreover, the functional linkages between soil microbiomes and grain quality attributes remain poorly elucidated. This study aims to address these gaps by systematically comparing conventional tillage with three ecological co-culture systems (rice-chicken, rice-fish, and rice-chicken-fish) across five critical rice growth stages. We hypothesize that co-culture systems enhance soil functionality and grain quality through stage-specific modifications of soil fertility, enzyme activities, and microbial communities, driven by synergistic plant–animal–microbe interactions. Specifically, this research seeks to answer the following questions: (1) What are the effects of different co-culture systems on soil properties across rice growth stages? (2) What are the effects of these systems on rice grain quality? (3) How do soil properties modulations influence grain quality? By employing a temporally explicit sampling design and integrative analysis, this study aims to elucidate the stage-specific mechanisms underlying soil–plant–microbe interactions in co-culture systems, providing a scientific basis for designing sustainable rice production systems based on precision ecological farming.

## 2. Materials and Methods

### 2.1. Experimental Design and Sampling

The experiment was conducted during the 2021 rice growing season at a specialized rice-poultry-fish integrated farm located in Changsha County, Hunan Province, China (113°13′09″ E, 28°24′48″ N; Figure 1A,B). The experimental soil was classified as reddish clayey paddy soil with the initial properties listed in Table 1.

The experimental components comprised rice cultivar Nongxiang 32, Partridge Shank chicken, and triploid hybrid crucian carp (*Carassius auratus* red var., ♀ × *Cyprinus carpio* ♂). Four experimental treatments were established, including conventional ridge-furrow paddy cultivation (CTL), rice-chicken (RC), rice-fish (RF), and rice-chicken-fish (RCF) co-culture systems. Each treatment was replicated three times, with individual replicate units consisting of a 60 cm^2^ paddy plot. Each plot included a cultivation ridge (12 m × 4.4 m, length × width) flanked by two drainage furrows (12 m × 0.3 m × 0.5 m, length × width × depth), providing space for rice cultivation and refuge for chickens and fish (Figure 1C). Adjacent plots were separated by alleys with plastic film barriers inserted into the soil.

Rice seedlings were sown in a seedbed on 25 May and transplanted on 21 June. Compound fertilizer (N:P_2_O_5_:K_2_O = 15:15:15) was applied as a basal fertilizer at 450 kg ha^−1^ and incorporated prior to transplanting. Seedlings were transplanted at a spacing of 25 cm × 15 cm, with four seedlings per hill. At 10 days after transplantation (DAT), experimental animals were introduced into designated co-culture systems, with 10 chickens (average weight 0.5 kg) per ridge plot (approximately 1500 chickens ha^−1^) for RC and RCF and 50 crucian carp (average length 8 cm, average weight 50 g) per furrow sector (approximately 8000 fish ha^−1^) for RF and RCF. Supplemental urea (46% N) was applied at 75 kg ha^−1^ once during the rice tillering stage. A foliar fertilizer containing potassium dihydrogen phosphate (KH_2_PO_4_, 5 g L^−1^) and urea (10 g L^−1^) was applied at 300 L ha^−1^ once during the booting stage and once during the early filling stage. Chickens and fish primarily foraged on naturally available resources (e.g., insects, zooplankton, and weeds). Supplemental feed was provided under scheduled, localized, and quantified protocols, with millet grain (1 kg per plot) for chickens and corn meal (0.2 kg per plot) for fish. No chemical pesticides or herbicides were applied throughout the experiment.

Chickens were removed from the ridges at the rice full heading stage to prevent grain damage. Fish harvesting and controlled drainage for field drying commenced one week prior to anticipated rice maturity. Rice was harvested on 28 September, threshed, dried, and stored for three months prior to quality analysis. Soil samples (0–20 cm plough layer) were collected from each plot using a standardized five-point sampling method at key rice phenological stages, including tillering (20 DAT), booting (40 DAT), heading (60 DAT), filling (75 DAT), and maturity (100 DAT). Samples were air-dried, ground, sieved, and stored at 4 °C for fertility and enzyme activity analysis. Post-harvest, additional soil samples (5–10 cm shallow layer) were collected similarly. After removing roots and impurities, these samples were homogenized and stored at −80 °C for microbial community analysis.

### 2.2. Determination of Soil Fertility Indicators

Soil fertility indicators (pH, OM, TN, TP, TK, AHN, AP, and AK) were analyzed using established protocols [23]. Soil pH was measured in a soil/distilled water suspension (2:5, *w*/*v*) using a calibrated pH meter (Mettler-Toledo™ FE20, Zurich, Switzerland) [23]. OM content was determined by the Walkley–Black potassium dichromate (K_2_Cr_2_O_7_) oxidation method [24]. TN and TP were quantified following concentrated sulfuric acid (H_2_SO_4_) digestion using a continuous flow analyzer (SEAL™ AA3, Norderstedt, Germany) [25]. TK and AK were extracted with sodium hydroxide (NaOH) and ammonium acetate (C_2_H_7_NO_2_), then measured by flame photometry [26]. AHN was assessed using the alkaline hydrolysis diffusion method [27]. AP was extracted with sodium bicarbonate (NaHCO_3_) and determined by UV-visible spectrophotometry [28].

### 2.3. Measurement of Soil Enzyme Activity

Soil catalase (CAT) activity was determined by potassium permanganate (KMnO_4_) titration. A mixture of 5 g soil, 0.5 mL toluene (C_7_H_8_), and 25 mL 3% hydrogen peroxide (H_2_O_2_) was shaken for 30 min. The reaction was terminated with 5 mL H_2_SO_4_. After filtration, 1 mL filtrate was reacted with 5 mL H_2_SO_4_ and titrated with KMnO_4_ until a faint pink endpoint appeared. Urease (UR) activity was measured colorimetrically using the sodium phenate-sodium hypochlorite (C_6_H_5_ONa-NaClO) method. A mixture of 5 g soil, 0.5 mL C_7_H_8_, 10 mL 10% urea solution, and 20 mL citrate buffer (pH 6.7) was incubated at 37 °C for 24 h. After filtration, 1 mL filtrate was reacted with 4 mL C_6_H_5_ONa and 3 mL NaClO for 20 min, and absorbance was measured at 578 nm. Invertase (INV) activity was assayed colorimetrically using 3,5-dinitrosalicylic acid (C_7_H_4_N_2_O_7_). A mixture of 5 g soil, 0.5 mL C_7_H_8_, 15 mL 8% sucrose solution, and 5 mL phosphate buffer (pH 5.5) was incubated at 37 °C for 24 h. After filtration, 1 mL filtrate was reacted with 3 mL DNS reagent and boiled for 5 min, and absorbance was measured at 540 nm.

### 2.4. Comparative Analysis of Soil Microbial Communities

Total genomic DNA was extracted from soil samples using the DNeasy^®^ PowerSoil^®^ Kit (QIAGEN, Hilden, Germany). Following verification of purity and quality, amplicon libraries targeting the V4 hypervariable region of the bacterial 16S rRNA gene were constructed. Paired-end sequencing was performed on an Illumina MiSeq platform (Allwegene Tech., Beijing, China). Raw paired-end reads were filtered using Trimmomatic (ver. 0.36) to eliminate adapter contamination and low-quality sequences with parameters SLIDINGWINDOW: 50:20 and MINLEN: 120. Overlapping pair-end reads were then merged using FLASH (ver. 1.20) with a minimum overlap of 10 bp. Chimeric sequences were identified and removed using Vsearch (ver. 2.7.1). The obtained clean reads were clustered into operational taxonomic units (OTUs) using the QIIME2 pipeline (ver. 2024.5) at 97% sequence similarity. Taxonomic classification was performed against the Silva database (Release 138) using the QIIME2 pipeline.

Venn and UpSet analysis were conducted using the R packages “VennDiagram” (ver. 1.7.3) and “UpSetR” (ver. 1.4.0). Alpha diversity, including observed species (Sobs) richness, Shanno diversity and Pielou evenness, beta diversity based on Bray–Curtis dissimilarity, and analysis of similarity (ANOSIM) were calculated using the packages “vegan” (ver. 2.7-1) and “picante” (ver. 1.8.2). Linear discriminant analysis effect size (LEfSe) analysis was performed using the package “microeco” (ver. 1.15.0) with a threshold of linear discriminant analysis (LDA) score ≥ 3 and *p* < 0.05. Co-occurrence network analysis was conducted using the packages “igraph” (ver. 2.1.4) and “psych” (ver. 2.5.6) and visualized in Gephi (ver. 0.10.1), considering correlations with |R^2^| > 0.8 and *p* < 0.05 (Pearson correlation with Benjamini-Hochberg correction). Neutral community models were fitted using the package “MicEco” (ver. 0.9.19). Functional potential was predicted via PICRUSt2 (ver. 2.5.2), and KEGG pathway was compared using STAMP (ver. 2.1.3). Mantel test, correlation analysis, and linear regression analysis were conducted using the packages “vegan” (ver. 2.7-1), “psych”, and “stats” (ver. 4.5.1), respectively.

### 2.5. Characterization of Rice Grain Quality

Paddy rice (300 g total, 100 g per replicate) was dehulled using a brown rice machine and milled with an inspection rice milling machine. Milled rice appearance quality (grain length, width, chalky grain rate, chalkiness degree) was assessed using a rice appearance quality scanner (Wseen™ SC-E, Hangzhou, China). According to the Chinese National Standard “GB/T 17891-2017 High Quality Pappy”, brown rice recovery, milled rice recovery, and head rice recovery were calculated [29]. Amylose content was determined by iodine blue colorimetry following the Chinese Agricultural Industry Standard “NY/T 2639-2014 Determination of Amylose Content in Rice—Spectrophotometry Method” [30]. Protein content was measured following concentrated H_2_SO_4_-NaOH digestion using a continuous flow analyzer (SEAL™ AA3, Norderstedt, Germany).

### 2.6. Statistical Analysis

Data were presented as mean ± standard deviation (SD). Significant differences among groups were assessed using one-way analysis of variance (ANOVA) followed by the Waller–Duncan post-hoc test in IBM SPSS Statistics (ver. 19.0). Statistical significance was defined as *p* < 0.05, with different lowercase letters denoting significant differences.

## 3. Results

### 3.1. Effects of Different Ecological Co-Culture Systems on Soil Fertility

Soil fertility indicators demonstrated complex and interconnected responses to ecological co-culture systems across key rice phenological stages (Table 2). Soil pH exhibited substantial fluctuations throughout the rice growth stages, while the CTL maintained significantly higher values at tillering (5.23 ± 0.04), heading (5.00 ± 0.02), and maturity (5.03 ± 0.01) stages compared to co-culture systems, particularly RC (tillering: 4.95 ± 0.01, heading: 4.41 ± 0.03, maturity: 4.47 ± 0.05) and RCF (tillering: 4.93 ± 0.07, heading: 4.45 ± 0.13, maturity: 4.91 ± 0.03). Conversely, at booting and filling stages, all co-culture systems exhibited higher pH than CTL (booting: 4.47 ± 0.22, filling: 4.92 ± 0.02), with RF demonstrating the highest values (booting: 5.06 ± 0.04, filling: 5.11 ± 0.01). OM content, crucial for nutrient retention, displayed temporal variation. Though CTL led at early stages (tillering: 22.91 ± 0.42 g kg^−1^, booting: 25.34 ± 0.57 g kg^−1^, heading: 27.86 ± 0.28 g kg^−1^), RCF surged to peak at maturity stage (28.82 ± 0.55 g kg^−1^), surpassing all other systems, while RC sustained consistently high OM levels throughout. AHN availability was significantly enhanced in RC throughout growth stages, recording the highest values (e.g., tillering: 143.67 ± 0.58 mg kg^−1^), significantly surpassing CTL, RF, and RCF at most stages. In contrast, RF generally displayed significantly lower AHN, particularly at booting stage (119.80 ± 2.11 mg kg^−1^), while CTL showed a significant decline in AHN from tillering (135.73 ± 7.48 mg kg^−1^) to maturity (120.00 ± 0.53 mg kg^−1^) stages. AP dynamics revealed that RC maintained significantly higher levels at booting (12.30 ± 0.70 mg kg^−1^) and heading (12.13 ± 0.26 mg kg^−1^) stages compared to other systems, although CTL showed significantly higher AP at filling (16.72 ± 0.26 mg kg^−1^) and maturity (16.44 ± 0.44 mg kg^−1^) stages. RF consistently recorded the lowest AP across nearly all stages. AK was markedly influenced by cropping pattern, with RC and RCF generally sustaining significantly higher levels during critical growth phases, particularly booting (RC: 100.67 ± 5.69 mg kg^−1^, RCF: 103.00 ± 6.08 mg kg^−1^) and filling (RC: 115.00 ± 2.00 mg kg^−1^, RCF: 110.00 ± 0.00 mg kg^−1^) stages. CTL displayed intermediate values, while RF consistently recorded the significantly lowest AK, especially at heading (60.00 ± 0.00 mg kg^−1^) and maturity (50.00 ± 0.00 mg kg^−1^) stages, indicating substantial depletion.

Critically, these shifts in available nutrients occurred despite limited variation in total nutrient pools. TN remained statistically similar in CTL (1.03 ± 0.25 g kg^−1^) and RC (1.04 ± 0.02 g kg^−1^), while RF (0.91 ± 0.03 g kg^−1^) and RCF (0.94 ± 0.01 g kg^−1^) showed significant reductions (Figure 2). TP was highest in RC (0.55 ± 0.01 g kg^−1^) but depressed in RCF (0.50 ± 0.02 g kg^−1^), and TK showed no significant treatment differences despite stark contrasts in AK availability (Figure 2), collectively indicating that cropping systems predominantly influenced nutrient mobilization processes rather than total reserves.

Collectively, these results demonstrated that soil fertility responses are highly dependent on both the specific cropping pattern implemented and the crop growth stage. Integrated systems such as rice-chicken and rice-chicken-fish co-cultures often showed higher available potassium levels but lower soil nitrogen levels compared to conventional tillage, whereas rice-fish co-culture maintained more stable pH values but frequently exhibited lower phosphorus and potassium levels.

### 3.2. Effects of Different Ecological Co-Culture Systems on Soil Enzyme Activities

Comprehensive analysis of soil enzymatic responses revealed distinct temporal patterns and system-specific effects on biochemical processes (Table 3). CAT activity progressively increased from tillering to heading stages in most systems, peaking at heading stage in RF (0.47 ± 0.01 mL g^−1^ 30 min^−1^), while RC maintained significantly higher activity during booting (0.45 ± 0.00 mL g^−1^ 30 min^−1^) and filling (0.44 ± 0.01 mL g^−1^ 30 min^−1^) stages compared to CTL. Notably, all systems exhibited significant suppression at maturity stage, with RF showing the most pronounced decline (0.31 ± 0.02 mL g^−1^ 30 min^−1^). UR dynamics revealed opposing temporal patterns, with CTL peaking at heading stage (0.32 ± 0.00 mg g^−1^ 24 h^−1^) and RF and RCF reaching maxima at filling stage (RF: 0.38 ± 0.02 mg g^−1^ 24 h^−1^, RCF: 0.38 ± 0.01 mg g^−1^ 24 h^−1^). RC showed significant inhibition at heading stage (0.16 ± 0.02 mg g^−1^ 24 h^−1^) despite initial elevation at tillering stage (0.30 ± 0.02 mg g^−1^ 24 h^−1^). INV activity demonstrated system-dependent divergence, with RC achieving the highest level at tillering (0.62 ± 0.01 mg g^−1^ 24 h^−1^) and booting (0.47 ± 0.02 mg g^−1^ 24 h^−1^) stages, contrasting with CTL, which showed significant suppression at booting stage (0.26 ± 0.05 mg g^−1^ 24 h^−1^) before rebounding to peak at heading stage (0.36 ± 0.01 mg g^−1^ 24 h^−1^). RF consistently yielded lower INV activity across later growth stages, particularly at maturity stage (0.32 ± 0.00 mg g^−1^ 24 h^−1^).

Collectively, these enzymatic dynamics revealed system-specific functional partitioning, demonstrating that cropping regimes stage-specifically regulate carbon mineralization, nitrogen transformation, and oxidative stress responses throughout the rice growth stages.

### 3.3. Effects of Different Ecological Co-Culture Systems on Soil Microbial Communities

#### 3.3.1. Soil Microbial Community Characterization and Diversity Profiling

A total of 2,132,444 high-quality reads were generated from 12 sequencing libraries, resulting in an average of 28,201 reads per sample. These sequences were clustered into OTUs, yielding 13,610 OTUs. OTU counts per sample ranged from 3735 (RF3) to 4582 (RF1), with a mean of 4190 (Figure 3A). Among all OTUs, 942, 822, 896, and 812 were unique to CTL, RC, RF, and RCF, respectively, while 3375 OTUs were shared among all systems (Figure 3B), indicating both commonality and system-specific microbial constituents.

Rarefaction curves plateaued, confirming sufficient sequencing depth to capture the majority of bacterial diversity (Figure 3C). Rank abundance curves exhibited moderate slopes, suggesting uniform distribution of bacteria taxa across samples (Figure 3C). Alpha diversity analysis revealed no significant differences among groups for Sobs, Shannon, or Pielou, indicating that ecological co-culture systems did not alter overall microbial richness or evenness (Figure 3D). In contrast, beta diversity assessed via non-metric multidimensional scaling (NMDS; Stress = 0.096) and ANOSIM demonstrated significant separation (R^2^ = 0.694, *p* < 0.001) among groups, underscoring that cropping system significantly influenced microbial community composition (Figure 3E).

#### 3.3.2. Microbial Community Composition and Differential Abundance Analysis

The soil bacterial community was dominated (>5% relative abundance) by Proteobacteria (CTL: 20.39%, RC: 17.44%, RF: 17.12%, RCF: 15.79%), Acidobacteriota (CTL: 15.87%, RC: 19.67%, RF: 15.32%, RCF: 17.57%), Desulfobacterota (CTL: 8.57%, RC: 7.58%, RF: 11.72%, RCF: 9.73%), Chloroflexi (CTL: 8.87%, RC: 9.71%, RF: 6.49%, RCF: 10.29%), Nitrospirota (CTL: 7.53%, RC: 8.19%, RF: 9.39%, RCF: 7.39%), and Verrucomicrobiota (CTL: 7.77%, RC: 6.97%, RF: 7.63%, RCF: 7.31%) at the phylum level (Figure 4A). Dominant genera (>1% relative abundance) included Unidentified (CTL: 36.39%, RC: 38.00%, RF: 38.72%, RCF: 38.39%), Uncultured (CTL: 11.88%, RC: 10.93%, RF: 11.96%, RCF: 11.54%), Uncultured bacterium (CTL: 10.72%, RC: 11.78%, RF: 11.25%, RCF: 11.42%), *ADurb.Bin063-1* (CTL: 2.10%, RC: 2.15%, RF: 1.80%, RCF: 2.12%), *Syntrophorhabdus* (CTL: 1.29%, RC: 1.21%, RF: 1.73%, RCF: 1.73%), and *Candidatus Solibacter* (CTL: 1.24%, RC: 1.65%, RF: 1.14%, RCF: 1.50%) (Figure 4A). LEfSe analysis identified system-specific biomarker taxa (LDA score ≥ 3, *p* < 0.05), showing that *MND1* and *Nitrospira* were enriched in CTL, with enrichment of *Rhodanobacter* and *Candidatus Solibacter* in RC, *Candidatus Nitrotoga*, *Desulfatiglans*, and *Syntrophus* in RF, and *Leptolinea* in RCF (Figure 4B). These results indicate that each co-culture system fostered a distinct microbial assemblage, likely driven by differences in organic inputs, animal activities, and resultant biochemical conditions.

#### 3.3.3. Microbiota Interactions, Community Assembly Mechanisms, and Functional Potential

Co-occurrence network analysis elucidated species interactions among groups (Figure 5A). The CTL network comprised 79 nodes and 100 edges, dominated by Proteobacteria. The RF network consisted of 88 nodes and 120 edges, dominated by Proteobacteria as well. The RC (82 nodes, 102 edges) and RCF (95 nodes, 153 edges) networks showed Actinobacteriota dominance. Co-culture systems exhibited higher average connectivity than the conventional tillage, indicating more complex and cooperative species interactions. The neutral community model explained a minor fraction (R^2^ < 0.3) of microbial community variation (Figure 5B), suggesting that community assembly was predominantly governed by deterministic processes. PICRUSt2 functional prediction identified 37 dominant KEGG pathways associated with soil microbiota. Remarkable functional divergence was observed, primarily in metabolism-related pathways between CTL and RC and in disease-associated pathways between CTL and RF/RCF (Figure 5C).

### 3.4. Effects of Different Ecological Co-Culture Systems on Rice Grain Quality

Distinct rice grain quality profiles emerged across ecological cropping systems, with integrated regimes demonstrating significant advantages in milling efficiency, grain appearance, and nutritional composition (Figure 6). RC achieved superior milling yields, evidenced by significantly higher brown rice recovery (BRR; 77.91 ± 3.02%) and milled rice recovery (MRR; 66.66 ± 0.72%) compared to CTL (BRR: 73.96 ± 2.25%, MRR: 64.94 ± 0.43%), while head rice recovery (HRR) remained statistically equivalent across all systems (Figure 6). Conversely, grain chalkiness metrics revealed substantial improvements under ecological management. CTL exhibited elevated chalkiness rate (15.23 ± 1.45%) and chalkiness degree (4.61 ± 0.71), whereas RC, RF, and RFC reduced these parameters by 20–30%. Kernel dimensions (length and width) showed no treatment-dependent variations. Nutritional analysis indicated consistent amylose content but divergent protein levels. RF (8.21 ± 0.25%) and RCF (8.20 ± 0.24%) enhanced protein content by 16.6% relative to CTL (7.04 ± 0.61%), whereas RC (7.25 ± 0.14%) reduced protein despite superior milling performance.

Collectively, these results delineated system-dependent compromises that RC optimizes processing quality but reduces nutritional value, while RF/RCF enhance nutrition with moderate milling efficiency. This highlighted the differential capacity of integrated ecological approaches to regulate specific grain quality attributes.

### 3.5. Correlations of Soil Fertility, Enzyme Activity, Microbial Community, and Grain Quality

To elucidate the influence of ecological co-culture systems on grain quality, associations among soil fertility parameters, enzyme activities, microbial communities, and grain quality attributes were investigated.

Mantel test identified pH, OM, AHN, AP, AK, TN, and INV as core functional modules through phenotypic matrix eigenvector decomposition and integration (Figure 7A). These variables exhibited significant correlations with microbial diversity and composition (r ≥ 0.4, *p* < 0.05; Figure 7A). Correlation heatmap analysis revealed contrasting relationships between INV and pH, with INV positively correlating with BRR and HRW, while pH demonstrated inverse patterns (Figure 7B). Furthermore, PC exhibited negative correlations with several soil fertility indicators, including OM, AP, and TN (Figure 7B). Notably, AHN emerged as a pivotal positive regulator of grain quality, showing significant positive correlations with beneficial traits, such as BRR, MRR, and HRW, while being negatively correlated with undesirable traits including CGR and CD (Figure 7B). Linear regression demonstrated that differentially abundant taxa were significantly associated with both enzyme activities and grain quality traits. Specifically, *Leptolinea* and *Desulfatiglans* showed negative correlations with CAT and INV activities, respectively, while *Rhodanobacter* and *Candidatus Solibacter* were positively correlated with INV (Figure 7C). Moreover, *Nitrospira* and *MND1* were negatively correlated with rice protein content, whereas *Syntrophus* and *Candidatus Nitrotoga* showed positive correlations (Figure 7D).

## 4. Discussion

This study provided comprehensive evidence that co-culture systems profoundly reshape soil functionality and rice grain quality through multifaceted linkages among soil fertility, enzyme activities, and microbial communities. Critically, these effects exhibited stage-dependent trajectories and system-specific functional differentiation, underscoring the multifunctional advantages of integrated farming over conventional monoculture.

Soil nutrients are fundamental to soil fertility, and integrated rice and aquatic animals systems have been shown to enhance nutrient accumulation [5,6,31]. Herein, pH fluctuations across phenological stages underscored the dynamic influence of co-cultures on soil acidity, potentially mediated through root exudate alterations, organic matter decomposition pathways, or animal waste inputs [32,33,34]. Specifically, RC, RF, and RCF generally reduced pH compared to CTL, consistent with previous studies [8,35]. This acidification is primarily attributed to nitrogen enrichment from high-protein feed residues and fecal deposition [8,35], compounded by reduced CO_2_ utilization due to animal foraging on plankton and weeds. Notably, RF increased soil pH, potentially due to enhanced buffering capacity from fish bioturbation and shifts in microbial communities favoring Actinobacteria [36]. The temporal dynamics of OM further revealed system-dependent effects. Although CTL accumulated OM earlier, RCF peaked OM at maturity (28.82 ± 0.55 g kg^−1^), suggesting improved late-season carbon retention. This aligned with the mechanism whereby animal excreta and root exudates stimulate microbial necromass accumulation [32,33,34]. Crucially, available nutrients (AHN, AP, and AK) exhibited temporal patterns decoupled from stable total nutrient pools (TN, TP, and TK), demonstrating that co-cultures primarily enhanced availability of key nutrients, particularly AK. This enhanced mobilization likely stems from animal bioturbation [37], manure-induced soil properties alterations [38], and microbial community modulation [39]. The significant depletion of AHN in CTL from tillering to maturity, contrasted with sustained levels in RC, further underscores the role of integrated systems in maintaining nitrogen availability.

Soil enzymes exhibited distinct temporal and system-driven patterns, reflecting differential regulation of carbon mineralization [40], nitrogen transformation [41], and oxidative stress responses [42]. The increase in CAT activity until heading, followed by a decline at maturity, aligns with patterns of reactive oxygen species generation during active plant growth and senescence [43,44,45]. Sustained elevated CAT activity in RF at heading (0.47 ± 0.01 mL g^−1^ 30 min^−1^) and filling (0.50 ± 0.03 mL g^−1^ 30 min^−1^) suggested enhanced oxidative stress mitigation during critical growth phases, potentially via increased microbial activity or organic inputs [46]. Antagonistic temporal patterns in UR and INV activities indicate functional niche partitioning among systems. Suppression UR in RC at heading (0.16 ± 0.02 mg g^−1^ 24 h^−1^) coinciding with peak AHN (130.60 ± 0.87 mg kg^−1^) suggests ammonium feedback inhibition [47,48] or pH-driven enzyme denaturation [41]. Conversely, delayed UR peaks in RF (0.38 ± 0.02 mg g^−1^ 24 h^−1^) and RCF (0.38 ± 0.01 mg g^−1^ 24 h^−1^) at filling indicate accelerated nitrogen mineralization, likely mediated by fish-induced microbial shifts. System-dependent INV dynamics further reflect distinct carbon allocation strategies, with RC showing early dominance and CTL peaking later, implying regulation of sucrose hydrolysis by carbon availability and microbial succession [49]. Collectively, these enzymatic responses demonstrate that co-culture systems differentially modulate biochemical processes governing carbon, nitrogen, and redox cycling across rice growth stages.

Soil microbiota play critical roles in promoting crop growth [50], driving nutrient cycling [51], enhancing stress tolerance [52], and sustaining agroecosystem functionality. Despite conserved alpha diversity, significant beta diversity separation (ANOSIM, R^2^ = 0.694, *p* < 0.001) confirmed that co-cultures restructured bacterial community. The dominance of core phyla (Proteobacteria, Acidobacteriota, Desulfobacterota, Chloroflexi, and Nitrospirota) is consistent with typical paddy soils [53,54]. The differential enrichment of biomarker taxa across cropping systems suggests niche-specific selection driven by system-specific physicochemical and biological alterations. For instance, the enrichment of *Nitrospira* in CTL may reflect a higher nitrification potential under conventional management, potentially leading to nitrogen loss through leaching or denitrification [55,56]. Conversely, the prevalence of *Syntrophus* in RF, known for its role in syntrophic fatty acid oxidation and nitrogen mineralization under anaerobic conditions, may enhance plant-available nitrogen, thereby contributing to the observed increase in grain protein content [57]. Similarly, the enrichment of *Candidatus Solibacter* in RC, a genus involved in metabolism and organic matter decomposition, may facilitate potassium mobilization, aligning with the sustained AK levels [58]. These taxon-specific responses underscore how co-cultures restructure microbial functional guilds to optimize nutrient cycling pathways. The poor fit of the neutral community model further reinforced deterministic processes, likely driven by system-specific differences in soil physicochemical properties, nutrient availability, and management practices, were primarily responsible for microbial community structuring. This finding aligns with our observation of significant beta diversity differences among treatments and reinforces the conclusion that ecological co-culture systems impose distinct selective pressures on soil microbiota, thereby enhancing functional specialization and ecosystem resilience. The increased network complexity and connectivity in co-cultures, particularly RCF (95 nodes, 153 edges), indicate enhanced microbial interactions and potential functional redundancy, which may confer greater metabolic versatility and nutrient cycling efficiency [59,60]. Functional predictions additionally revealed significant divergence in metabolic and disease-associated pathways, implying specialized functional consequences for soil properties and plant-microbe interactions. These findings collectively suggest that ecological co-culture systems foster a more interactive and functionally diversified microbiome, which supports nutrient turnover and system stability.

The system-specific modulation of soil properties translated into distinct grain quality profiles. RC improved milling efficiency (BRR: 77.91 ± 3.02%, MRR: 66.66 ± 0.72%), likely attributable to consistently enhanced AK levels that promote grain filling and structural integrity [61,62]. Conversely, RF and RCF significantly increased grain protein content (RF: 8.21 ± 0.25%, RCF: 8.20 ± 0.24%) despite lower AHN, suggesting fish-mediated nitrogen cycling through bioturbation and excreta inputs [63]. Reduced chalkiness across all co-cultures indicates improved grain quality, potentially due to balanced nutrient supply and abiotic stress mitigation mediated by improved soil biological activity and pH stability. Correlation analysis established direct soil-grain quality linkages, with identified core functional modules (pH, OM, AHN, AP, AK, TN, INV) significantly correlating with microbial metrics, indicating that these factors collectively shape the soil microenvironment and influence microbial community structure and function. The contrasting relationships of INV activity and pH with milling yields implicated sucrose hydrolysis and soil acidity as key regulators of processing quality [64]. Conversely, the negative correlations between PC and soil nutrients (e.g., OM, AP, and TN) may reflect nitrogen immobilization or microbial nutrient competition in organically enriched soils, thereby limiting nitrogen allocation to grains. The pronounced role of AHN as a pivotal regulator of grain quality highlighted its critical function in improving nitrogen availability during grain filling, thus promoting both yield and quality formation. Crucially, associations of specific biomarker taxa with both enzyme activities and protein content directly implicated cropping pattern-induced microbial shifts as determinants of nutritional quality, primarily through their mediation of nitrogen transformations governing plant uptake and assimilation. For instance, the negative correlations of *Leptolinea* and *Desulfatiglans* with CAT and INV activities, respectively, suggest their potential roles in modulating oxidative stress and carbon metabolism, whereas the positive associations of *Rhodanobacter* and *Candidatus Solibacter* with INV indicate their contribution to sucrose decomposition [58]. Most notably, the negative relationships of *Nitrospira* and *MND1* with protein content align with their known roles in nitrification, which may lead to nitrogen loss through leaching or denitrification, whereas the positive correlations of *Syntrophus* and *Candidatus Nitrotoga* with protein content suggest their involvement in anaerobic nitrogen mineralization, enhancing plant-available nitrogen and thereby supporting protein synthesis in grains [55,56,57]. These findings collectively emphasize that co-cultures regulate grain quality through complex interactions within the soil–microbe–plant continuum, driven by specific biochemical processes and microbial functional guilds.

## 5. Conclusions

This study present a comprehensive assessment of the modifications caused by rice-based coculture systems (RC, RF, and RCF) on soil properties and grain quality relative to conventional tillage (CTL). Our results demonstrate that co-cultures induce stage-specific and system-dependent shifts in soil fertility, enzyme activities, and microbial communities, ultimately improving targeted aspects of rice grain quality. The crucial novel insights include the following: (1) The RC system consistently enhanced AK levels without altering TK pools and optimized milling recovery, underscoring the role of avian activity (e.g., defecating, burrowing, and foraging) in nutrient mobilization and grain filling. (2) RF and RCF systems maintained AHN levels despite reduced TN pools, peaked UR activity at filling stage, and elevated grain protein content, highlighting fish-mediated nutrient cycling and availability through bioturbation and excreta. (3) Microbial community structures were system-specific reconstructed, with increased network complexity and connectivity in co-cultures, indicating enhanced functional redundancy and ecological resilience. These results demonstrate that ecological co-culture systems optimize agroecosystem functionality through tailored soil–plant–microbe interactions, reducing reliance on synthetic inputs while improving grain quality. Integrative analysis linked specific microbial taxa (e.g., *Syntrophus* and *Nitrospira*) to quality traits, providing a predictive framework for designing precision agroecological practices. This work advances the field by delineating temporal dynamics and mechanistic pathways often overlooked in static comparisons, offering science-based strategies for sustainable rice production systems.

## Figures and Tables

**Figure 1 biology-14-01195-f001:**
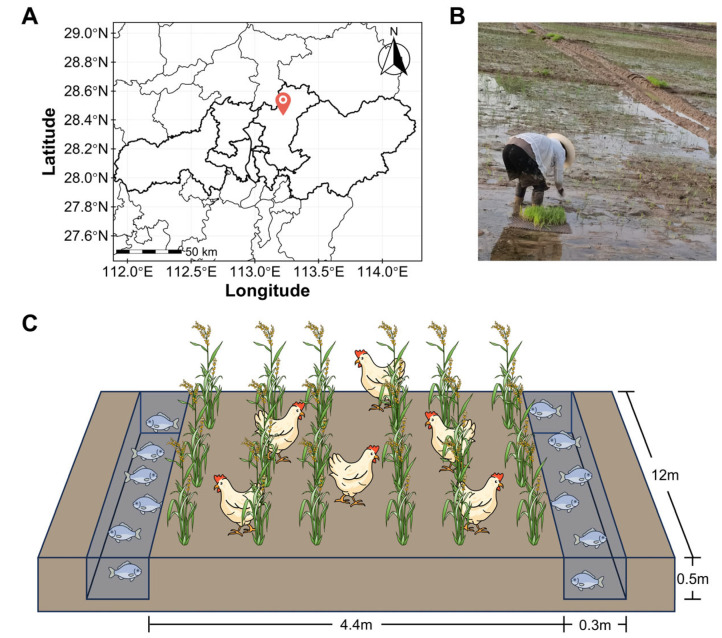
Experimental design and site characterization. (**A**) Georeferenced map of the study area; (**B**) In situ photographic documentation of field conditions; (**C**) Schematic representation of the rice-chicken-fish co-culture integrated farming system.

**Figure 2 biology-14-01195-f002:**
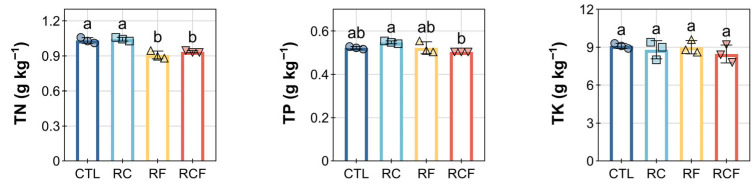
Effects of different ecological co-culture systems on soil total nitrogen (TN), total phosphorus (TP), and total potassium (TK) content at the maturity stage. Distinct symbols (circles, square, triangles, and inverted triangles) represent different groups, and lowercase letters denote significant differences (ANOVA, Waller–Duncan post-hoc test, *p* < 0.05).

**Figure 3 biology-14-01195-f003:**
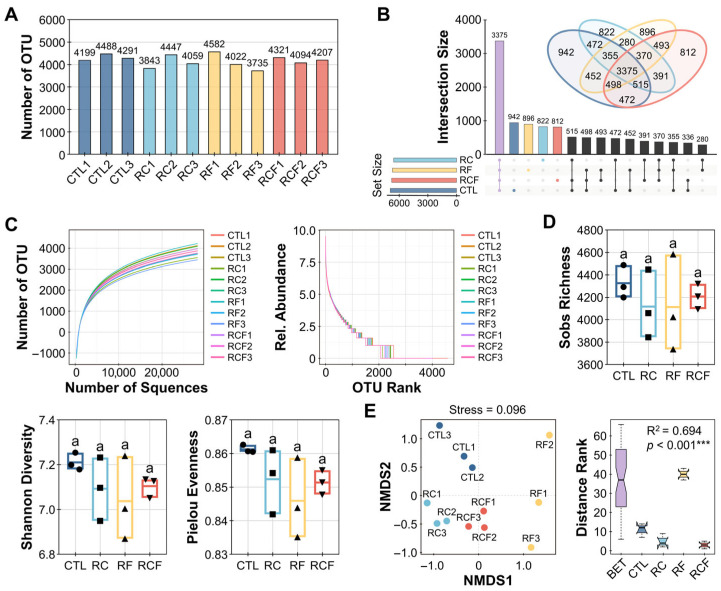
Soil microbial community characterization and diversity profiling. (**A**) Operational taxonomic unit (OTU) quantification per sample; (**B**) Venn and Upset plots identifying exclusive/shared OTUs among groups; (**C**) Rarefaction curves (**left panel**), constructed at a 97% sequence similarity cut-off, and rank abundance curves (**right panel**), illustrating sequencing depth and species distribution; (**D**) Inter-group alpha diversity comparisons; Distinct symbols (circles, square, triangles, and inverted triangles) represent different groups, and lowercase letters denote significant differences (ANOVA, Waller–Duncan post-hoc test, *p* < 0.05); (**E**) Non-metric multidimensional scaling (NMDS) plot based on Bray–Curtis matrices visualizing beta-diversity separation (**left panel**) with analysis of similarity (ANOSIM) validation of group differences (**right panel**).

**Figure 4 biology-14-01195-f004:**
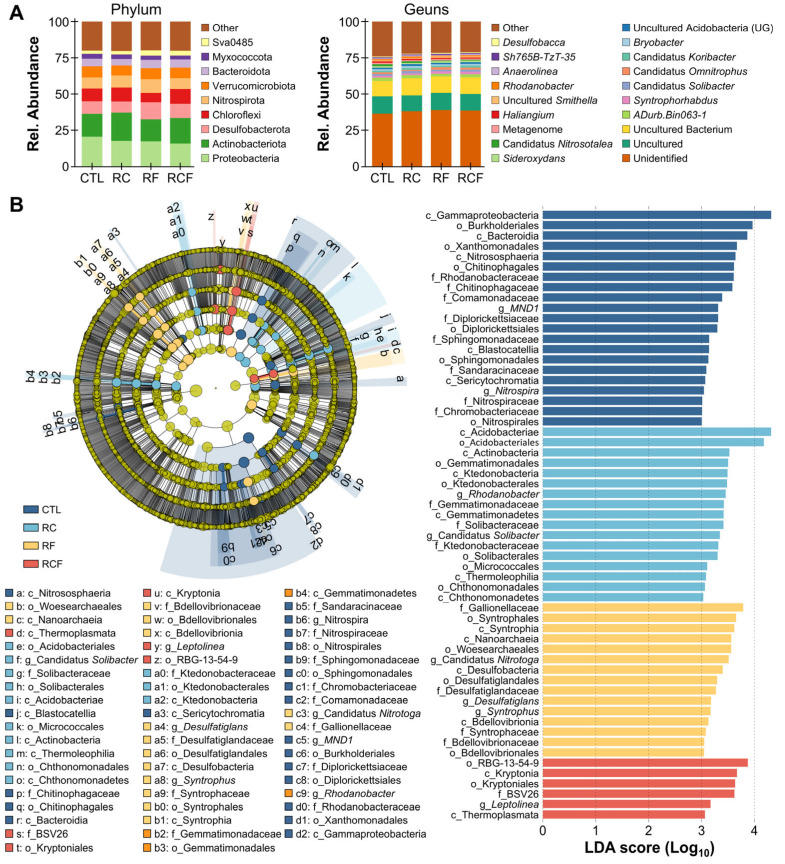
Microbial community composition and differential abundance analysis. (**A**) Taxonomic profiling depicting relative abundances at the phylum (left panel) and genus (right panel) levels among groups; (**B**) Identification of distinct taxa among groups derived from linear discriminant analysis effect size (LEfSe) analysis with a threshold of linear discriminant analysis (LDA) score ≥ 3 and *p* < 0.05.

**Figure 5 biology-14-01195-f005:**
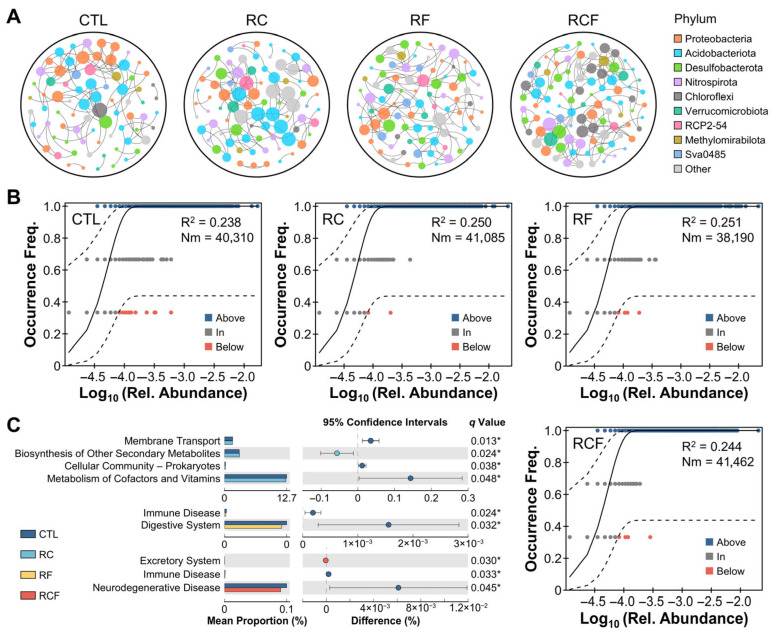
Microbiota interactions, community assembly mechanisms, and functional potential. (**A**) Network analysis depicted co-occurrence patterns of soil microbiota at the phylum level; (**B**) Neutral community model evaluating stochastic assembly processes; (**C**) Functionally predicted KEGG pathways differing among groups; Statistical significance was set at * *p* < 0.05 (Welch’s *t*-test, FDR multiple test correction). Note: KEGG pathway annotations referred to evolutionarily conserved functional orthologs present in soil microbiota and were not indicative of human clinical conditions.

**Figure 6 biology-14-01195-f006:**
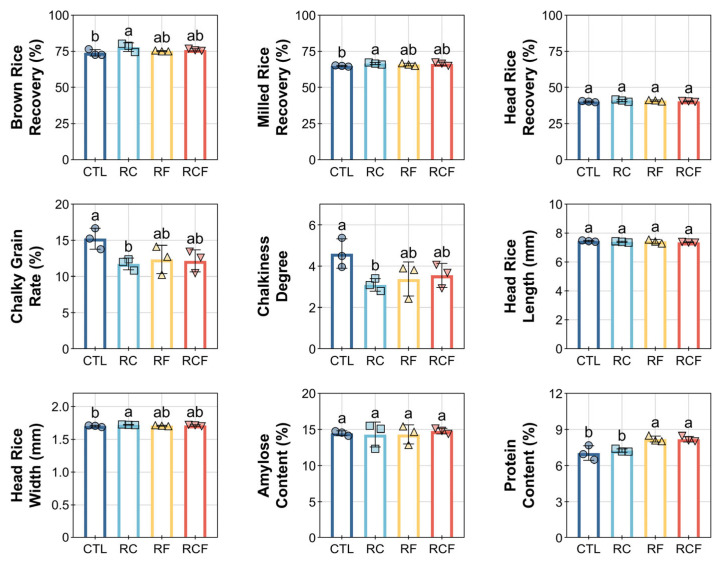
Effects of different ecological co-culture systems on rice grain quality. Distinct symbols (circles, square, triangles, and inverted triangles) represent different groups, and lowercase letters denote significant differences (ANOVA, Waller–Duncan post-hoc test, *p* < 0.05).

**Figure 7 biology-14-01195-f007:**
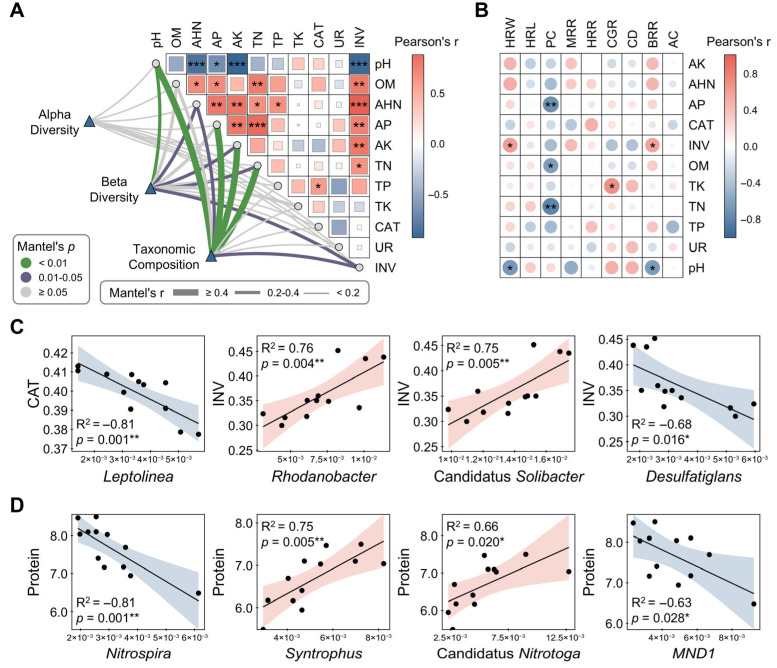
Integrative analysis of soil properties and grain quality relationships. (**A**) Mantel test evaluating associations between microbial communities and soil fertility parameters; (**B**) Heatmap illustrating pairwise Pearson correlations between soil fertility indices and grain quality traits; (**C**) Linear regression modeling relationships between differential taxa and enzyme activities; (**D**) Linear regression modeling relationships between differential taxa and key grain quality metrics. Statistical significance was set at * *p* < 0.05, ** *p* < 0.01, and *** *p* < 0.001 (Pearson correlation). Abbreviations: OM, organic matter; AHN, alkali-hydrolyzable nitrogen, AP, available phosphorus; AK, Available Potassium; TN, total nitrogen; TP, total phosphorus; TK, total potassium; CAT, catalase; UR, urease; INV, invertase; BRR, brown rice recovery; MRR, milled rice recovery; HRR, head rice recovery; CGR, chalky grain rate; CD, chalkiness degree; HRL, head rice length; HRW, head rice width; AC, amylose content; PC, protein content.

**Table 1 biology-14-01195-t001:** Initial properties of the experimental soil.

Parameters	Value
Organic Matter (OM)	26.32 g kg^−1^
Total Nitrogen (TN)	0.79 g kg^−1^
Total Phosphorus (TP)	0.51 g kg^−1^
Total Potassium (TK)	7.87 g kg^−1^
Alkali-Hydrolyzable Nitrogen (AHN)	133.70 mg kg^−1^
Available Phosphorus (AP)	13.28 mg kg^−1^
Available Potassium (AK)	103.33 mg kg^−1^

Data were presented as mean ± standard deviation (SD).

**Table 2 biology-14-01195-t002:** Effects of different ecological co-culture systems on soil fertility.

Parameters	Group	Tillering Stage	Booting Stage	Heading Stage	Filling Stage	Maturity Stage
pH	CTL	5.23 ± 0.04 ^a^	4.74 ± 0.22 ^b^	5.00 ± 0.02 ^a^	4.92 ± 0.02 ^b^	5.03 ± 0.01 ^a^
	RC	4.95 ± 0.01 ^b^	4.91 ± 0.01 ^ab^	4.41 ± 0.03 ^b^	4.95 ± 0.10 ^b^	4.47 ± 0.05 ^c^
	RF	4.98 ± 0.10 ^b^	5.06 ± 0.04 ^a^	4.96 ± 0.08 ^a^	5.11 ± 0.01 ^a^	5.02 ± 0.07 ^a^
	RCF	4.93 ± 0.07 ^b^	4.99 ± 0.06 ^ab^	4.45 ± 0.13 ^b^	4.94 ± 0.01 ^b^	4.91 ± 0.03 ^b^
OM (g kg^−1^)	CTL	22.91 ± 0.42 ^a^	25.34 ± 0.57 ^a^	27.86 ± 0.28 ^a^	23.69 ± 0.64 ^b^	26.10 ± 0.34 ^c^
	RC	22.17 ± 0.52 ^a^	24.56 ± 0.37 ^a^	27.77 ± 0.42 ^a^	27.22 ± 0.32 ^a^	27.72 ± 0.28 ^b^
	RF	22.77 ± 0.28 ^a^	22.31 ± 0.81 ^b^	25.89 ± 0.57 ^b^	24.69 ± 0.83 ^b^	27.03 ± 0.55 ^bc^
	RCF	20.15 ± 0.36 ^b^	22.59 ± 0.42 ^b^	25.84 ± 0.29 ^b^	24.25 ± 1.46 ^b^	28.82 ± 0.55 ^a^
AHN (mg kg^−1^)	CTL	135.73 ± 7.48 ^ab^	131.07 ± 2.95 ^b^	125.30 ± 1.67 ^b^	127.53 ± 0.97 ^b^	120.00 ± 0.53 ^c^
	RC	143.67 ± 0.58 ^a^	135.53 ± 1.47 ^a^	130.60 ± 0.87 ^a^	135.40 ± 0.87 ^a^	135.07 ± 2.10 ^a^
	RF	131.53 ± 3.74 ^b^	119.80 ± 2.11 ^c^	123.47 ± 2.14 ^b^	125.67 ± 2.86 ^b^	121.20 ± 5.70 ^bc^
	RCF	129.33 ± 0.76 ^b^	134.27 ± 0.23 ^ab^	125.67 ± 1.51 ^b^	124.87 ± 2.23 ^b^	127.73 ± 0.95 ^b^
AP (mg kg^−1^)	CTL	11.91 ± 0.44 ^a^	8.83 ± 0.39 ^c^	8.60 ± 0.59 ^b^	16.72 ± 0.26 ^a^	16.44 ± 0.44 ^a^
	RC	12.24 ± 0.50 ^a^	12.30 ± 0.70 ^a^	12.13 ± 0.26 ^a^	14.26 ± 0.84 ^b^	14.71 ± 0.54 ^b^
	RF	9.39 ± 0.17 ^b^	9.84 ± 0.26 ^b^	8.38 ± 0.17 ^b^	10.34 ± 0.35 ^d^	10.84 ± 0.59 ^d^
	RCF	12.47 ± 0.51 ^a^	9.61 ± 0.48 ^bc^	8.88 ± 0.17 ^b^	13.03 ± 0.35 ^c^	13.70 ± 0.26 ^c^
AK (mg kg^−1^)	CTL	105.33 ± 3.21 ^b^	54.67 ± 5.51 ^b^	80.00 ± 0.00 ^b^	90.00 ± 0.00 ^b^	90.00 ± 0.00 ^b^
	RC	122.00 ± 6.08 ^a^	100.67 ± 5.69 ^a^	103.33 ± 5.77 ^a^	115.00 ± 2.00 ^a^	96.00 ± 2.00 ^a^
	RF	90.00 ± 1.00 ^c^	60.67 ± 1.53 ^b^	60.00 ± 0.00 ^c^	63.33 ± 5.77 ^c^	50.00 ± 0.00 ^d^
	RCF	129.67 ± 4.04 ^a^	103.00 ± 6.08 ^a^	81.00 ± 1.00 ^b^	110.00 ± 0.00 ^a^	80.00 ± 0.00 ^c^

Data were presented as mean ± SD. Lowercase letters denote significant differences (ANOVA, Waller–Duncan post-hoc test, *p* < 0.05).

**Table 3 biology-14-01195-t003:** Effects of different ecological co-culture systems on soil enzyme activities.

Parameters	Group	Tillering Stage	Booting Stage	Heading Stage	Filling Stage	Maturity Stage
Catalase Activity (CAT; mL g^−1^ 30 min^−1^)	CTL	0.35 ± 0.00 ^a^	0.41 ± 0.02 ^b^	0.45 ± 0.01 ^bc^	0.39 ± 0.02 ^c^	0.41 ± 0.00 ^a^
	RC	0.33 ± 0.02 ^ab^	0.45 ± 0.00 ^a^	0.46 ± 0.01 ^ab^	0.44 ± 0.01 ^b^	0.35 ± 0.02 ^b^
	RF	0.35 ± 0.01 ^ab^	0.41 ± 0.01 ^bc^	0.47 ± 0.01 ^a^	0.50 ± 0.03 ^a^	0.31 ± 0.02 ^c^
	RCF	0.32 ± 0.01 ^b^	0.40 ± 0.01 ^c^	0.44 ± 0.01 ^c^	0.42 ± 0.03 ^bc^	0.34 ± 0.01 ^b^
Urease Activity (UR; mg g^−1^ 24 h^−1^)	CTL	0.23 ± 0.01 ^a^	0.26 ± 0.01 ^a^	0.32 ± 0.00 ^a^	0.29 ± 0.02 ^c^	0.20 ± 0.01 ^ab^
	RC	0.30 ± 0.02 ^b^	0.26 ± 0.00 ^a^	0.16 ± 0.02 ^d^	0.33 ± 0.01 ^b^	0.19 ± 0.02 ^ab^
	RF	0.28 ± 0.02 ^a^	0.18 ± 0.03 ^c^	0.20 ± 0.00 ^c^	0.38 ± 0.02 ^a^	0.17 ± 0.02 ^b^
	RCF	0.25 ± 0.01 ^b^	0.24 ± 0.01 ^b^	0.26 ± 0.00 ^b^	0.38 ± 0.01 ^a^	0.21 ± 0.01 ^a^
Invertase Activity (INV; mg g^−1^ 24 h^−1^)	CTL	0.39 ± 0.02 ^c^	0.26 ± 0.05 ^c^	0.36 ± 0.01 ^a^	0.30 ± 0.04 ^b^	0.38 ± 0.02 ^a^
	RC	0.62 ± 0.01 ^a^	0.47 ± 0.02 ^a^	0.28 ± 0.00 ^b^	0.44 ± 0.04 ^a^	0.40 ± 0.02 ^a^
	RF	0.37 ± 0.04 ^c^	0.32 ± 0.02 ^bc^	0.24 ± 0.03 ^c^	0.31 ± 0.02 ^b^	0.32 ± 0.00 ^b^
	RCF	0.47 ± 0.00 ^b^	0.37 ± 0.01 ^b^	0.25 ± 0.01 ^c^	0.36 ± 0.01 ^b^	0.30 ± 0.01 ^b^

Data were presented as mean ± SD. Lowercase letters denote significant differences (ANOVA, Waller–Duncan post-hoc test, *p* < 0.05).

## Data Availability

Data is contained within the article.

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
