# Peer review of "Beyond Monoculture: A Comparative Analysis of Soil Properties and Grain Quality in Rice-Based Co-Culture Systems"

_biology, 2025, doi:10.3390/biology14091195_

Round 1
Reviewer 1 Report
Comments and Suggestions for Authors
- The title is comprehensive but too long and complex.
-
Simplify and clarify sentence structures throughout manuscript.
-
Strengthen balance by briefly addressing limitations of co-culture systems in introduction section and Condense objectives into sharper research questions.
-
Research gap is missing in introduction.
-
How your study is differ from others.
-
What is novelty. Please add in manuscript.
-
What hypothesis behind this study.Methodology:
-
Experimental Design and Sampling-Chicken and fish stocking densities are given, but it would be helpful to express them in per hectare terms for comparability with other studies.
- Provide reference citations for each analytical method (e.g., Walkley-Black for OM, Kjeldahl or modified Kjeldahl for TN).
- Important soil enzymes like dehydrogenase, phosphatase, and β-glucosidase are not included. Provide more detail on incubation conditions (time, temperature, soil-to-solution ratio). Without this, reproducibility is limited.
- Comparative Analysis of Soil Microbial Communities:
-
State the sequencing depth (average reads per sample, cut-off for rarefaction). This is critical for assessing diversity analyses.
-
Clarify quality filtering criteria (e.g., minimum Phred score, sequence length cut-off).
-
Consider mentioning whether archaea and fungi were excluded intentionally, as the focus is entirely on bacterial 16S.
-
Co-occurrence network analysis: specify whether Spearman or Pearson correlations were used, and clarify how multiple testing was controlled (e.g., FDR correction).
-
Neutral community models: briefly explain why they were chosen and what ecological insights they are expected to provide.
- Figure 2 quality is not good.
- The discussion currently summarizes results but does not sufficiently interpret mechanisms or connect findings with broader literature.
- Rewrite conclusion. It contain general statement.
- The conclusion is overly general and repetitive, restating findings without highlighting novelty or applied significance.
Reviewer 2 Report
Comments and Suggestions for Authors
- The term "CAT" first appears in the abstract at line 53 without an explanation of its full meaning. It is recommended to provide an explanation here or avoid using abbreviations in the abstract;
- The abstract lacks a clear statement of the novelty or innovation of the research and experimental data to support it. It is recommended to highlight the unique aspects of the article in the abstract and add experimental data to emphasize the experimental results;
- In Section 2.1, it is recommended to present the initial soil property data in tabular form for greater clarity;
- The content in lines 277–279 appears to be subjective. Why are some changes described as “increased availability” (positive) while others are described as “decreased content” (negative)?
- In Fig. 3, the labels for "OTU" and "OUT" are inconsistent and need to be uniformly corrected;
- Figs. 4, 5, and 7 present a large amount of data, but there is no deeper analysis of these data, and there is a lack of in-depth interpretation of the causes or mechanisms behind the data differences;
- The discussion section lacks data comparisons to support its analysis. Additionally, the article does not delve deeply into the roles of microbial communities and enzymes or their impacts. It is recommended to explore these aspects further and explicitly link microscopic-level findings (e.g., microbial changes, enzyme activity) with macroscopic-level observations (e.g., soil properties, grain quality);
- Key conclusions rely heavily on citations from existing literature rather than emphasizing the unique findings of this study. It is recommended to use your own data to support certain arguments and clearly articulate the consistency and differences between the findings of this study and previous conclusions in the literature. Some paragraphs read like repetitive restatements of results.
Reviewer 3 Report
Comments and Suggestions for Authors
This a comprehensive study, systematically conducted, and well-written. Just a few minor suggestions:
Page 2, line 48: from “…16S rRNA-based microbial profiling, and (milling recovery, …” to “…16S rRNA-based microbial profiling, and rice grain quality (milling recovery, …”
Page 2, line 53: from “…de-layed …” to “…delayed …”
Page 2, line 58: from “… proper-ties …” to “… properties …”
Page 3, lines 71-73: Meaning is not clear; please rephrase.
Page 4, line 140: Do you mean bioavailable P and K?
Page 6, lines 201-202: What tool did you use to process raw reads to OTU? Did you use QIIME2 for this task? You did not get ASVs but OTUs? Please clarify.
Page 12, Fig. C: How do pathways related to immune diseases, digestive system, and neurodegenerative diseases impact human health?
